# Spherical Wrist Manipulator Local Planner for Redundant Tasks in Collaborative Environments

**DOI:** 10.3390/s23020677

**Published:** 2023-01-06

**Authors:** Marcello Chiurazzi, Joan Ortega Alcaide, Alessandro Diodato, Arianna Menciassi, Gastone Ciuti

**Affiliations:** 1The BioRobotics Institute, Scuola Superiore Sant’Anna, 56127 Pisa, Italy; 2Department of Excellence in Robotics and AI, Scuola Superiore Sant’Anna, 56127 Pisa, Italy

**Keywords:** robotics control, local path planner, task redundancy, collision avoidance strategy, human–robot interaction

## Abstract

Standard industrial robotic manipulators use well-established high performing technologies. However, such manipulators do not guarantee a safe Human–Robot Interaction (HRI), limiting their usage in industrial and medical applications. This paper proposes a novel local path planner for spherical wrist manipulators to control the execution of tasks where the manipulator number of joints is redundant. Such redundancy is used to optimize robot motion and dexterity. We present an intuitive parametrization of the end-effector (EE) angular motion, which decouples the rotation of the third joint of the wrist from the rest of the angular motions. Manipulator EE motion is controlled through a decentralized linear system with closed-loop architecture. The local planner integrates a novel collision avoidance strategy based on a potential repulsive vector applied to the EE. Contrary to classic potential field approaches, the collision avoidance algorithm considers the entire manipulator surface, enhancing human safety. The local path planner is simulated in three generic scenarios: (i) following a periodic reference, (ii) a random sequence of step signal references, and (iii) avoiding instantly introduced obstacles. Time and frequency domain analysis demonstrated that the developed planner, aside from better parametrizing redundant tasks, is capable of successfully executing the simulated paths (max error = 0.25°) and avoiding obstacles.

## 1. Introduction

Industrial robotic manipulators use well-established technologies that guarantee high performances in terms of velocity and accuracy. Indeed, typical values of repeatability and joint velocity for industrial manipulators are approximately 0.01 mm and 300°/s, respectively [1]. Despite the clear capabilities of industrial robots, their usage is still very much limited to applications where their workspace is exclusive and humans’ presence is forbidden. The standard regulatory ISO 10218-1/2:2011 [2] and ISO/TS 15066:2016 [3] provide requirements, specifications, and guidelines to ensure safety for industrial and collaborative robotic applications and work environments. Nowadays, most industrial robots work inside fences that physically limit the access of humans into the workspace. However, certain tasks such as robotic surgery require the robot to share its workspace with humans. The need for physically close interaction between humans and machines has motivated many groups, both in academia and industry, to study new strategies for safe Human–Robot Interaction (HRI) [4,5]; the followed strategies for a safe HRI can be categorized as post-contact or pre-contact measures.

Post-contact safety measures serve to mitigate the effects of a collision once it has already occurred. Post-contact measures include soft and smooth designs of any potential contact points on the robot structure and the use of integrated sensing capabilities measuring the intensity of such impacts [6,7]; post-contact measures are used to minimize the impact of a collision but also intentional contacts [8,9].

On the other hand, pre-contact strategies use perceptive systems to provide online environment information to the robotic manipulator to generate collision-free trajectories. Path planning algorithms can be classified into: (i) global path planning algorithms, and (ii) local path planning algorithms. Global path planning algorithms solve an optimization problem searching for a free-collision path from an initial configuration to the desired one [10]. Global planners are usually able to find the optimal path in a finite amount of time; however, their computational time requirements can be a limiting factor in dynamic environments. On the other hand, local planners provide a local trajectory to the robotic arm based on the final goal and the local environment information at a certain time. Local planners require much less computational time to be executed and generally cannot guarantee that the generated trajectories will reach the final goal. Local planners are therefore adequate for tasks characterized by having a highly changing objective trajectory (e.g., teleoperation) or highly dynamic environment (e.g., collaboration with humans). Most of the local path planners in literature are based on and/or are inspired by the artificial potential field foundational method introduced by Khatib et al. in [11]. The artificial potential field method assigns a repulsive potential to the obstacles and an attractive potential to the desired goal configuration. Potential field-based methods have proved to effectively avoid collisions in real-time applications [12].

Robotic manipulators executing tasks in non-structured, dynamic environments must guarantee both a safe interaction with the environment and a safe execution of the objective task, especially in critical tasks such as the ones performed by surgical robots. Many such tasks are characterized by having an axis of redundancy aligned with the last joint of the robotic manipulator. Assuming that basic machining operations, such as milling and drilling, only require 5-DoF, the anthropomorphic robot becomes adequate and the task optimizable. The optimization method can be used to exploit the redundancy that certain tasks have. Singularities, joint limits, and collisions were optimized for redundant manipulators [13]. Lukić et al. proposed the optimization of the Cartesian stiffness of a kinematic redundant robot with a null space projection. However, they only considered maintaining position without any specified orientation, which is not the case for machining operations [14]. Several research groups have studied approaches to exploit redundancy in industrial applications. Zanchettin et al. [15] implemented a redundancy resolution criterion that maximizes the manipulator maneuverability to exploit the redundant degree of freedom available on drilling tasks. In [16], Guo et al. presented a novel method based on the Jacobian matrix for computing a performance index based on the stiffness of the robot during machining applications. These studies also paved the way for the use of robotic arms in redundant tasks related to medical scenarios, such as in Focused Ultrasound Surgery [17] or for teleoperation control of a 7-DoF robot manipulator for Minimally Invasive Surgery (MIS) [18].

The manipulation needs, found in the application described in [17], have motivated the development of the novel local planner presented in this work. In this specific robotic scenario, an anthropomorphic manipulator (i.e., an anthropomorphic arm with a spherical wrist) is equipped with a transducer able to stimulate human tissue through ultrasound energy for treating tumors in moving organs. The ultrasound energy is concentrated in a focal spot located along the central axis of the transducer. Hence, the pose of this rotation axis, combined with the manipulator end-effector (EE) linear position, determines the focal spot positioning (i.e., 5-DoF task due to symmetrical tool). The remaining DoF can be used to accomplish secondary tasks such as cable management. Similarly, there are many other scenarios where the position of the joint $J_{t}$ does not affect the main task, as in welding applications. Hence, it would be useful to describe the EE angular movement decoupling the rotation of the third wrist joint $J_{t}$ from the rest of the angular motions.

### Aim and Organization of the Work

This work aims to develop a local planner that optimizes and simplifies the safe usage of robotic manipulators equipped with a spherical wrist executing redundant tasks in workspaces shared with dynamic obstacles (e.g., humans). The proposed path planner is based on the following features: (i) the rotation of the third wrist joint $J_{t}$ must be decoupled from the rest of the EE angular motion, (ii) the resulting EE manipulator dynamics should behave as a linear dynamical system, (iii) the collision avoidance strategy must consider the entire surface of the manipulator, and (iv) all the local planner parameters must have a physical meaning.

The present work has four main sections. In Section 2, we first present the theoretical formulation that leads to the local planner and then describe the methods used to validate the proposed local planner making use of an ad hoc simulator. In Section 3, we present and discuss the results of the performed simulations. Finally, we summarize the conclusions in Section 4.

## 2. Materials and Methods

### 2.1. Local Planning for Redundant Collaborative Tasks: Theoretical Formulation

In this subsection, we present a new parametrization of the pose of the end-effector in a spherical wrist manipulator that more naturally represents the fundamental degrees of freedom of redundant tasks. The parametrization decouples the rotation of the third wrist joint $J_{t}$ from the rest of the EE angular motion. We then present the theoretical computation of the disturbance vector D_EE_ based on the collision avoidance strategy. Finally, we present the control law used to implement the local path planner.

#### 2.1.1. Task Parametrization: Separation of the Redundant Axis

The kinematic model of the robotic manipulator depends on the structure of the manipulator [19]. The spherical wrist represents one of the most widely employed joint configurations and its structure is presented in Figure 1. Figure 1 also represents the manipulator base and the manipulator EE reference system following the Denavit–Hartenberg convention [19]. This robotic structure has two important properties: (i) the pose of the $z_{EE}$ axis depends only on the joints before $J_{t}$ and (ii) the rotation along $z_{EE}$ can be independently controlled by means of the $J_{t}$. These properties do not depend on the entire robotic structure, but they are intrinsic proprieties of the spherical wrist. We propose a parametrization that separates the EE angular movement into two rotations: (i) a rotation along the axis perpendicular to both the z-axis of the initial orientation, $z_{s}$, and the target orientation, $z_{t}$, and (ii) a rotation of the joint $J_{t}$ to the desired rotation along the z-axis.

In Figure 2, we present a generic motion of the z-axis together with the described parameters and rotation angles (i.e., θ and γ) along their respective axes of rotation. Figure 2b,c illustrate the parallel and perpendicular views concerning the plane defined by the $z_{s}$ and $z_{t}$ vectors. Figure 2b presents the θ angle rotation along the $x_{\theta }$ axis perpendicular to the $z_{s}$ and $z_{t}$ plane. On the other hand, the rotation along the $y_{\gamma }$ axis performs the out-of-plane rotation. The $y_{\gamma }$ axis is defined to be perpendicular to the $x_{\theta }$ axis and to the projection of $z_{EE}^{\theta }$ into the plane spanned by the θ angle. In the following equations, we formalize the described definitions of the rotation axes $x_{\theta }$ and $y_{\gamma }$.
(1)$$x_{\theta }=z_{s}\times z_{t}$$
(2)$$y_{\gamma }=z_{EE}^{\theta }\times x_{\theta }$$

We can then compute the $z_{EE}$ at any instant applying the following ordered rotations to the starting z-axis ($z_{s}$).
(3)$$z_{EE}=R_{y_{\gamma }}\left(\gamma \right)R_{x_{\theta }}\left(\theta \right)z_{s}$$

The notation $R_{a}\left(b\right)$ refers to the rotation along the a-axis by an angle *b*. Based on the definitions of the rotation axis we will use the equation 3 to express the angular motion of the EE; the following expression can be used to compute the reference angle $\theta_{r}$ (Equation (3)) while, by definition, the reference angle $\gamma_{r}$ is always zero.
(4)$$\theta_{r}=\cos^{-1}\left(z_{s}^{T}z_{t}\right)$$

More specifically, starting from a random $z_{s}$ and using Equations (1)–(3), we can compute the $z_{EE}$ by multiplying $z_{s}$ for two different rotation matrices. A first rotation is performed along $x_{\theta },$ which is the axes and the angle needed to align $z_{s}$ and $z_{t}$ (target orientation). A second rotation is performed along $y_{\gamma }$, which is the axes (perpendicular to the plane containing $x_{\theta }$ and the projection of $z_{EE}$) and the angle needed to align $z_{t}$ with $z_{EE}$.

Note that the definition of $x_{\theta }$ guarantees that $\theta_{r}$ is always larger than or equal to zero. Given a measured EE orientation, such orientation can be expressed in the described parametrization by applying the following equation where $\gamma_{r}$ is the reference target angle (i.e., the angle needed to reach the target orientation), whereas γ is the state variable that evolves (i.e., the real angle).
(5)$$\gamma_{r}=\sin^{-1}\left(x_{\theta }^{T}z_{EE}\right)$$
(6)$$\bar{\theta }=\cos^{-1}\left(z_{s}^{T}z_{EE}^{\theta }\right)$$

Finally, we compute the last joint $J_{t}$ directly applying the manipulator inverse kinematics. It is worth noting that the angle position qt of the joint $J_{t}$ does not influence the z-axis overlapping motion.

##### Singularity Handling

The described rotation axis is not well defined when $z_{s}$ and $z_{t}$ are parallel. The $x_{\theta }$ axis can be chosen as an arbitrary vector contained in the mutually perpendicular plane. If the scalar product $z_{s}\;\cdot \;z_{t}$ = −1, an EE rotation is requested to achieve the desired orientation. This freedom of choice can be used to avoid robotic wrist singularity [19]. Hence, setting the $x_{\theta }$ axis as the rotation axis of $\mathrm{the}\;\mathrm{joint}\;J_{f}$ allows performing the EE rotation through only its joint angle. By doing so, the second joint $J_{s}$ does not perform any movement, thus allowing it to avoid the robotic wrist singularity.

#### 2.1.2. Disturbance Computation for Collision Avoidance

Inspired by the artificial potential field method [11], we propose to introduce a disturbance vector modifying the planned trajectory based on the distance information between obstacles $O_{i}$, i = 1, …, N, and each manipulator link. Each obstacle contributes to such a disturbance vector introducing a virtual force $F_{EE_{i}}$ and one virtual torque $T_{EE_{i}}.$ For each obstacle $O_{i}$, we define the distance between an obstacle and the manipulator as the minimum distance between the obstacle and each of the M manipulator links. We then compute the virtual force $F_{EE_{i}}$ generated by the obstacles using the following piece-wise function.
(7)$$\left\{\begin{array}{c} F_{EE_{i}}=F_{MAX}-\frac{\left(F_{MAX}-F_{W}\right)}{d_{D}}d_{O_{i}}^{m};\;\;\;\;\;\mathrm{if}\;(||d_{O_{i}}^{m}||<d_{D})\\ F_{EE_{i}}=F_{W}-\frac{F_{W}}{\left(d_{W}-d_{D}\right)}\left(d_{O_{i}}^{m}-d_{D}\right);\;\;\;\;\;\mathrm{if}\;(d_{D}<||d_{O_{i}}^{m}||<d_{W})\\ 0;\;\;\;\mathrm{if}\;(||d_{O_{i}}^{m}||>d_{W}) \end{array}\right.$$

In Figure 3, we present the two linear zones: the” warning” and” danger” zones resulting from the proposed Equation (7). In these zones, each obstacle acts as linear springs with different stiffness (i.e., greater stiffness in the” danger” zone). Four parameters characterize the function: (i) the starting distance of the” warning” zone $d_{W}$, (ii) the starting distance of the “danger” zone $d_{D}$, (iii) the obstacle force F_W_ generated at $d_{W}$, and (iv) the maximum obstacle force $F_{MAX}$ generated at zero distance.

We use the virtual forces computed to generate a virtual torque. We compute the virtual torque as the direct sum of the following two torque components: (i) the torque perpendicular to the $z_{EE}$ axis ($T_{EE_{i}}^{p}$) and (ii) the torque along the $z_{EE}$ axis ($T_{EE_{i}}^{a}$). The virtual torque $T_{EE_{i}}^{a}$ is non-zero only when the minimum distance $d_{O_{i}}^{m}$ is associated with the last manipulator link. The following equation presents the computation of the virtual torque $T_{EE_{i}}^{a}$ based on the force $F_{EE_{i}}$ vectors components perpendicular to the $z_{EE}$ axis and the application lever arm, normalized with the maximum lever arm (i.e., $\frac{T_{l}}{2},\;T_{l}$ being the thickness of the last manipulator link).
(8)$$T_{EE_{i}}^{a}=\frac{2}{T_{l}}\left(\left(I_{3x3}-z_{EE}z_{EE}^{T}\right)\left(p_{O_{i}}-EE\right)\right)\times \left(I_{3x3}-z_{EE}z_{EE}^{T}\right)F_{{\mathrm{EE}}_{i}}$$

We compute the virtual torque $T_{EE_{i}}^{p}$ perpendicular to the z_EE_ axis as the cross-product between F_EEi_ and the normalized lever arm along the $z_{EE}.$
(9)$$T_{EE_{i}}^{p}=\frac{1}{\left||z_{EE}^{T}\left(EE-W\right)|\right|}\left(z_{EE}z_{EE}^{T}\left(p_{O_{i}}-EE\right)\right)\times F_{EE_{i}}$$

Finally, we sum the force and torque contribution of each obstacle obtaining the overall virtual force and torque.
(10)$$F_{EE}=\underset{i}{\sum }F_{EE_{i}}\;$$
(11)$$T_{EE}=\underset{i}{\sum }T_{EE_{i}^{a}}+\underset{i}{\sum }T_{EE_{i}^{p}}\;$$

Once we have the virtual torque $T_{EE}$ computed in the base manipulator frame, we can express it using the transformation that we present below.
(12)$$T_{EE}^{s}=\left[\begin{array}{c} \begin{array}{c} x_{\theta }^{T}\\ y_{\gamma }^{T}\\ z_{EE}^{T} \end{array} \end{array}\right]T_{EE}$$

If we compose the virtual force and torque expressed by the Cartesian and custom axis, respectively, we obtain the disturbance vector $D_{EE}$.
(13)$$D_{EE}=\left[\begin{array}{c} F_{EE}\\ T_{EE}^{s} \end{array}\right]$$

It is worth noting that other virtual forces/torque may be superimposed based on contact forces/torques to implement an impedance/admittance control. The measurement of the actual contact force/torque can be provided by external sensors, such as the sensitive and protective skin presented in [6,20] and/or standard load cells.

#### 2.1.3. Control Law

The state vector **X**, defined below, represents the manipulator EE pose and it is composed of the Cartesian coordinates of the EE and the angle:(14)$$X={\left[x\;y\;z\;\theta \;\gamma \;q_{t}\right]}^{T}$$

The decoupled nature of the state variables allows using a decentralized Multiple Input Multiple Output (MIMO) linear dynamical system to control the dynamics of **X**. Each state variable is controlled through a Single Input Single Output (SISO) system with a closed-loop architecture. Figure 4 depicts the structure of the SISO control system (i.e., equal for the six state variables) and it introduces the state vector reference $X_{r}$ and disturbance vector $D_{EE}$.

Whenever the disturbance and velocities are not saturated, the system behaves linearly. Under those conditions, we can use the superimposition principle to write the transfer function from the inputs $X_{r}$ and $D_{EE}$ to the output X, as follows:
(15)$$X=\frac{1}{\frac{D}{K}s+1}X_{r}+\frac{\frac{1}{K}}{\frac{D}{K}s+1}D_{EE}=F\left(s\right)X_{r}+\frac{1}{K}F\left(s\right)D_{EE}$$

The transfer function F(s) controls the dynamics of the state vector, parametrized through the damping *D* and spring *K* parameters (i.e., the pole of the closed-loop system is $\frac{D}{K}$). Both systems inputs equally influence the **X** dynamics with different static gains: 1 for the reference $X_{r}$ and $\frac{1}{K}$ for the disturbance $D_{EE}$. The SISO control system includes two saturations: (i) one saturates the state variable velocity and (ii) the other saturates the maximum amplitude of the disturbance. We define the following linear velocity saturation function.
(16)$$\left[\begin{array}{c} \dot{x_{s}}\\ {\dot{y}}_{s}\\ {\dot{z}}_{s} \end{array}\right]=\frac{S_{l}}{\left||{\left[\dot{x\;}\dot{y\;}\dot{z}\right]}^{T}|\right|}\left[\begin{array}{c} \dot{x}\\ \dot{y}\\ \dot{z} \end{array}\right]$$

Similarly, we define the following saturation function for the EE angular velocity *ω*.
(17)$$\omega =\left[\begin{array}{ccc} x_{\theta } & y_{\gamma } & z_{EE} \end{array}\right]\left[\begin{array}{c} \dot{\theta }\\ \dot{y}\\ \dot{z} \end{array}\right]$$
(18)$$\left[\begin{array}{c} \dot{\theta_{s}}\\ {\dot{\gamma }}_{s}\\ {\dot{q}}_{t_{s}} \end{array}\right]={\left[\begin{array}{ccc} x_{\theta } & y_{\gamma } & z_{EE} \end{array}\right]}^{-1}(S_{a}\frac{\omega }{\left||\omega |\right|})$$

We used just two saturation parameters to saturate the EE velocity to maintain the motion direction unchanged, following industry standards [1]. On the other hand, we may use individual saturation constants for each state variable of the disturbance signal.

### 2.2. Validation Methodology of the Theoretical Formulation: Simulations

In this subsection, we present the architecture, the simulation environment, and the tests that we used to validate the proposed local planner. A fundamental element of the simulator is the collision and proximity simulator (CPS). The CPS was developed and used by the authors in [21]. The local planner simulator parameters, used to validate the local planner, are presented in the following subsection, together with the representative simulated scenarios.

#### 2.2.1. Simulator

##### Simulator Architecture

In Figure 5, we present the general architecture of the proposed local path planner. When a new desired EE pose is provided to the path planner, the reference block computes the state variable’s reference $X_{r}$ and the axes $x_{\theta }$ and $y_{\gamma }$. The reference angle $\theta_{r}$ is calculated using Equation (3), while the references for $q_{t_{r}}$ are computed using the manipulator inverse kinematics. The linear state variables do not need computation because they directly correspond to EE Cartesian coordinates. The inputs of the MIMO controller are the state variable reference $X_{r}$, the $D_{EE}$ disturbance vector, and the measure of the state variables $\bar{X}$ provided by the block f_2_. The measure of the EE Cartesian coordinate and the joint angle **q_t_** are directly provided by the manipulator controller. The $D_{EE}$ disturbance vector is computed based on the CPS information, as presented in the following subsection. The input of the disturbance block is the current manipulator joints angle necessary to bring the virtual manipulator in the simulator. The output of the MIMO block is the state vector **X**, which is transformed into a manipulator joint trajectory $q_{ref}$ by means of f_1_. Block f_1_ uses the manipulator inverse kinematics to compute the references of the joints before $J_{t}$ is obtained from **X**. The reference signal to the manipulator $q_{ref}$ is obtained by adding the state variable q_t_ to the previously calculated joint reference.

##### Collision and Proximity Simulator

The collision and proximity simulator is written in C++ language and is based on the Bullet Physics engine [22], following the performance analysis conducted in [23]. The software architecture of the simulator is based on client–server architecture; thus, different applications can interface with the CPS through a simple dedicated interface (e.g., socket applications). To reduce the computational time for collisions and proximity algorithm detection, we simplified the geometries of the manipulator links. The use of a simplified version of the manipulator links is a common practice in the development of collision simulators [24]. The working environment is displayed through an open-source viewer that can be turned off to reduce the CPS computational time. Given a set manipulator pose, the CPS outputs a list of information related to each manipulator link. In particular, the CPS output reports the minimum distance $d_{O_{i}}$ (with point of application $p_{O_{i}}$ and vector $v_{O_{i}}$) between each manipulator link and each obstacle in the virtual working environment.

##### Simulator Parameters

In this work, we use a model of an ABB IRB120 (Zurich, Switzerland) as a representative example of an industrial manipulator with a spherical wrist. The simulator runs in Matlab (MathWorks Inc., Natick, MA, USA) and interfaces with our CPS software [20]. The model of the manipulator is equipped with a tool with maximum thickness, $T_{l}$, of 92.8 mm. The dynamics of the manipulator actuator are neglected in the performed simulation. The local planner requires two tuning steps: (i) the tuning of the MIMO system, which controls the dynamics of the manipulator, and (ii) the tuning of the collision avoidance strategy. The MIMO system is defined by the following parameters: (i) the spring parameter *K*, (ii) the damping parameter D, and (iii) the velocity and disturbance saturation functions. The spring parameter *K* is set to 1 to not amplify or attenuate the disturbance **D_EE_**. The desired linear manipulator dynamics are set to have a settling time (5%) of 3 s (i.e., a pole of the closed loop at 1 rad/s). The pole of the linear system is controlled by the ratio between *K* and *D*; therefore, the damping parameter *D* is set to 1. The saturation thresholds for the linear and angular velocities are set to 100 mm/s and 10°/s, respectively. The disturbance saturation for the linear state variable is set to 100 N, whereas 30 N/mm is set for the angular state variable. These settings lead to 100 mm and 30° of maximum displacement from $X_{r}$, since the spring parameter *K* is 1 when obstacles appear in the workspace. The collision avoidance strategy is only defined by the parameters of Equation (7), which determine the virtual forces generated by obstacles. The parameters set in the simulations are 300 and 100 mm for the $d_{w}$ and $d_{D}$, respectively, while the forces are set to 25 N (*F_W_*) and 100 N (*F_MAX_*).

#### 2.2.2. Validation Scenarios

Three generic scenarios have been simulated using the previously described simulator. In the first scenario, the local planner is used to follow a periodic signal as would happen, for example, for a medical robot that needs to adapt its motion to the human breath. Secondly, the local planner is requested to follow a random sequence of step signals that could represent, for example, a set of motions required for welding or teleoperating a robot. Finally, a set of scenarios are simulated to validate the suitability of the local planner to avoid collision with obstacles. It is worth noting that given the linear nature of the local planner, the superimposition principle can be applied to separately investigate the manipulator’s EE response to the reference signal $X_{ref}$ and the disturbance $D_{EE}$. The robot references generated by the local planner in all cases have been analyzed both in the time and frequency domains. The time domain analysis compares the EE trajectories with the related reference and nominal signals. The nominal signals are computed by exciting the nominal linear system (i.e., F(s)) with the related reference signals. The EE linear motion analysis is performed only by studying the state variable x. This is possible because the Cartesian coordinates are decoupled. On the other hand, angular motion requires a complete EE angular movement investigation (i.e., roll, pitch, and qt angles). The frequency analysis is performed by comparing the spectrum of the joint reference $q_{ref}$ with the spectrum of nominal dynamics (which uses the RPY parametrization). This serves to investigate how the inverse kinematics affect the joint reference $q_{ref}$ spectrum, also evaluating its suitability for standard manipulator actuator joints. All the simulations are performed ensuring no velocity saturation occurs.

##### Periodic Signal

The reference signals used for the periodic signal following the scenario are generated using the roll, pitch, and yaw (RPY) parametrization performed in local axes. The 6-DoF sinusoidal trajectories are composed of three harmonic frequencies (i.e., 0.4, 0.2, and 0.1 rad/s), both for the linear and angular coordinates. We have used the following reference trajectories for the EE Cartesian position and orientation.
(19)$$\left[\begin{array}{c} \begin{array}{c} EE_{r}^{x}\\ EE_{r}^{y}\\ EE_{r}^{z} \end{array} \end{array}\right]=50\left[\begin{array}{c} \begin{array}{c} \sin\left(0.4t\right)\\ \sin\left(0.2t\right)\\ \sin\left(0.1t\right) \end{array} \end{array}\right]+\left[\begin{array}{c} \begin{array}{c} 300\\ 0\\ 200 \end{array} \end{array}\right]\;$$
(20)$$\left[\begin{array}{c} R_{r}\\ P_{r}\\ q_{t_{r}} \end{array}\right]=25\left[\begin{array}{c} \begin{array}{c} \sin\left(0.4t\right)\\ \sin\left(0.2t\right)\\ \sin\left(0.1t\right) \end{array} \end{array}\right]\;$$

##### Sequence of Step Signals

A sequence of 100 6-DoF step reference signals has been randomly generated using the roll, pitch, and yaw (RPY) parametrization in local axes. It is worth noting that the roll and pitch angles define the pose of the target axis $z_{t}$, whereas the yaw angle is directly related to the angle position qt of the joint $J_{t}$ [18]. The EE Cartesian position for the 6-DoF step reference is randomly generated in a cube with 200 mm side centered.

Figure 6 reports angular motion from an initial to the desired orientation of the manipulator’s EE (depicted in blue). In red, the time evolution of the EE orientation sampled at 0.1 s is reported. The inset represents the θ dynamical response for the depicted angular motion at [300, 0, 200]^T^ mm, whereas the EE orientations are randomly generated using RPY parametrization with a maximum amplitude of 25°.

##### Close-Obstacle Collision Avoidance

The collision avoidance strategy is assessed by inserting obstacles (represented as spheres) in the workspace without varying the EE reference position ([300, 0, 300]^T^ mm with zero RPY angles). Six different configurations of obstacles are chosen as the case studies. The first three simulations include a single obstacle positioned at different $z_{b}$ coordinates (i.e., 350, 450, and 550 mm) with $x_{b}$ = 300 mm and $y_{b}$ = −150 mm. The other simulations include multiple obstacles (i.e., 2 and 4 spheres) to assess the superposition of the proposed collision avoidance strategy. Two simulations are performed with two obstacles: one has the obstacles on the same side of the manipulator, $y_{b}$ = −150 mm) and the other has the obstacles on opposite sides, one in $y_{b}$ = −150 mm and one in $y_{b}$ = 150 mm). The last simulation is performed in a symmetric configuration with four obstacles placed around the manipulator at different $z_{b}$ coordinates.

## 3. Results

### 3.1. Time Domain Analysis Results

In Figure 7*,* we report the evolution of x, the nominal dynamics, and the relative step reference signal $x_{r}$. As can be observed, the dynamics of x follow the nominal dynamical response with the tuned settling time. Indeed, the maximum error between the real and nominal dynamics is negligible (i.e., ∼6–10 mm). Figure 6 shows the angular motion trajectory (in red), sampled at 10 Hz, from a starting orientation to a target orientation (in blue). In the absence of any disturbances, the motion of the $z_{EE}$ axis evolves along the plane defined by the $z_{s}$ and $z_{t}$ vectors, driven by the state variable *θ*. As can be seen in Figure 6, the *θ* dynamics are equal to the desired and nominal dynamics tuned with the *K* and *D* parameters. The angle position qt belongs to **X;** thus, it evolves identically to the nominal response. Hence, the linear and angular EE motions are linked to a linear dynamical system with the imposed settling time for step references. Figure 8a,b report the dynamic evolution of the coordinates x and qt, their nominal dynamics, and the relative signals reference (see Equations (19) and (20)). The roll and pitch dynamics are reported in Figure 8c,d, respectively. The graphs present an almost perfect match between the nominal and real dynamics; indeed, the errors between them are 0.08 and 0.25° for roll and pitch angles, respectively.

### 3.2. Frequency Domain Analysis Results

The results of the frequency analysis for step and sinusoidal paths are reported in Figure 9a,b, respectively. The graphs report the mean and the maximum spectrum of the joint’s references $q_{ref}$ and the mean spectrum of the nominal dynamics (parametrized with RPY angle). Figure 9a shows that the mean and maximum $q_{ref}$ spectra are very similar to the spectrum of the nominal dynamics. Therefore, the non-linearity introduced by the manipulator inverse kinematics does not significantly affect the $q_{ref}$ spectrum. On the other hand, differences between the nominal spectrum and $q_{ref}$ spectrum can be observed for the sinusoidal path. The manipulator inverse kinematics introduces some components multiple of the exciting input frequencies (i.e., ultra-harmonic frequencies), highlighted in Figure 9b. Nevertheless, the non-desired ultra-harmonic frequencies are attenuated after the linear system band-pass, becoming negligible with the increase in frequency. Indeed, the maximum value of the $q_{ref}$ spectrum is 19.87 (100%), whereas the maximum values after 1 rad/s and 1 Hz are 0.23 (1.16 %) and 0.02 (0.10 %), respectively. Therefore, the dynamics of the joint reference $q_{ref}$ are suitable for typical robotic manipulator actuator joints, since its band-pass is larger than 1 Hz [1,25].

### 3.3. Close-Obstacle Collision Avoidance

Figure 10 report the initial poses of the validation scenarios. The $x_{b}$ coordinate is fixed to 300 mm for all the simulated obstacles, as can be observed in Figure 10a. This choice allows us to describe the EE movements through just two of the state variables (y and γ) without loss of generality. Indeed, the forces produced by the obstacles are principally exerted along $y_{b}$. These forces cause a torque along the $x_{b}$ axis because the lever arm is mainly along $z_{b}$. Subject to this potential field, the resultant EE motion is mainly a translation along $y_{b}$ and a rotation along $x_{b}$. We have decided to describe this rotation with the state variable γ, responsible for the collision avoidance strategy. The results presented in Figure 10 show how the value of γ increases and the value of y decreases when the obstacle z_B_ coordinate increases. As could be expected, the local planner responds to the configuration presented in Figure 10d, where the obstacle is closer to the manipulator’s wrist than to the manipulator EE, by separating the wrist from the obstacle and keeping the EE close to the Cartesian reference position (i.e., $y_{r}$ = 0).

Figure 11 illustrates the resulting dynamics of the state variables y, γ, q_t_, and the time response of the minimum distance for the three performed simulations with one obstacle. Figure 11a,b present that the fastest change of the state variables happens when the obstacle is at 550 mm due to the initial obstacle distance (i.e., 58 mm versus 85 mm, as highlighted in Figure 11d). During the initial phase, the distance constantly increases because the angular disturbance saturation is active (the torque is 50.09 N/m at 1 s on state variable γ). The resulting response allows us to conclude that the state variable’s responses can be esteemed as a saturated linear dynamical response with different steady-state values. Figure 11c shows that the state variable q_t_ remains zero for the obstacle at 550 mm given that the obstacle is closer to the penultimate manipulator link than to the last manipulator link. Figure 12 reports the final poses of the validation scenarios. Figure 12a reports the results of the simulations with two obstacles on the same side of the manipulator. The final value of the state variables y and γ is larger than those achieved in the one-obstacle simulations, as expected given the addictive nature of the collision avoidance strategy. The results of the simulation with obstacles on different sides of the manipulator are presented in Figure 12b, resulting in a negative y due to the proximity of the lower obstacle to the EE. Finally, the results of the four obstacles simulation presented in Figure 12c show no significant manipulator motion, as can be expected from the symmetrical obstacle configuration.

## 4. Conclusions

The described local planner provides a more natural way of describing 5-DoF tasks by using a parametrization of the EE orientation that decouples the rotation of the third wrist joint $J_{t}$ from the rest of the EE angular motion. The developed parametrization represents the physical behavior of the manipulator in a decoupled manner, facilitating both the interpretation and the tuning of the control parameters. Indeed, θ represents the in-plane rotation along the minimal path, the *γ* angle represents the out-of-plane rotation, and qt is the position of the third joint of the spherical wrist. The proposed local planner, based on a decentralized MIMO linear system with closed-loop architecture, has been demonstrated to allow the imposition of the EE dynamics to behave as a first-order linear system, facilitating any desired tuning of the EE dynamic response. Approaches using robot redundancy allow us to solve and locally optimize the robot path planning in a dynamic non-structured environment where the manipulator employs potential field approaches. In this regard, the presented approach enables industrial and medical applications where robot stiffness and dexterity can greatly improve task efficiency. It is worth noting that the proposed parametrization can be easily adapted to control 7-DoF manipulators by adding the elbow angle introduced in [26,27] into the state vector **X**. Additionally, the proposed local planner integrates a custom collision avoidance strategy that has proven to successfully deform the reference trajectory to maintain the manipulator separated from surrounding obstacles. The proposed collision avoidance strategy has proven to enhance human safety with a computationally efficient and simple-to-tune disturbance vector that does not require setting control points onto the robotic manipulator.

## Figures and Tables

**Figure 1 sensors-23-00677-f001:**
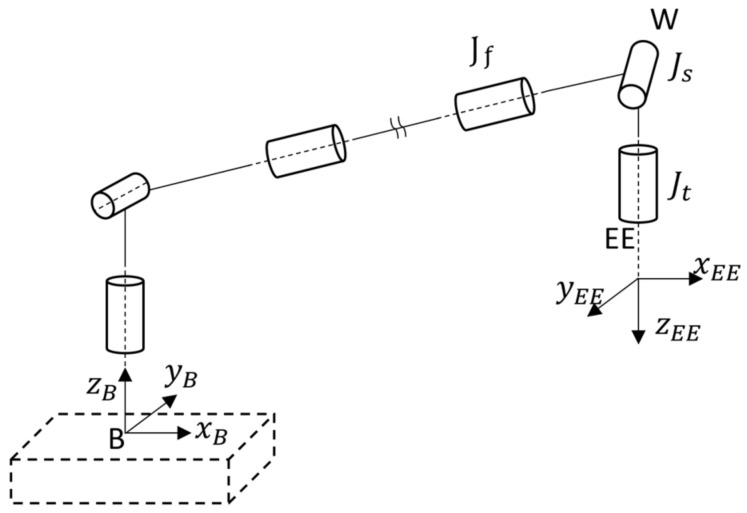
Kinematic model of a 6-DoF manipulator with focus on the spherical wrist, where the center of the spherical wrist (W) and the end-effector (EE) reference system are presented; a possible manipulator base reference system (B) is also represented.

**Figure 2 sensors-23-00677-f002:**
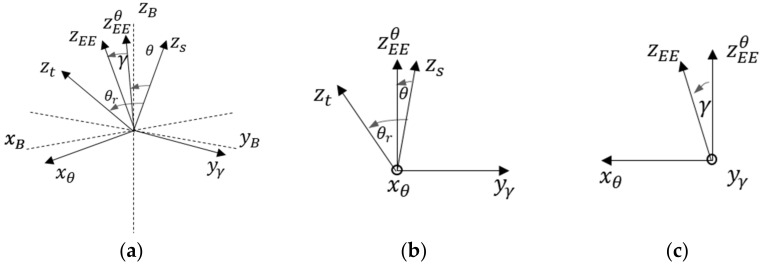
(**a**) Schematic representation of the z-axis overlapping motion for reaching the vector z_t_ starting from z_s_. The angles θ and γ are defined relative to the rotation axis x_θ_ and y_γ_. (**b**) Section of the plane defined by the vectors z_t_ and z_s_ spanned by the θ angle. (**c**) Section of the plane spanned by the γ angle perpendicular to the x_θ_ and $z_{\mathrm{EE}}^{\theta }$ vector.

**Figure 3 sensors-23-00677-f003:**
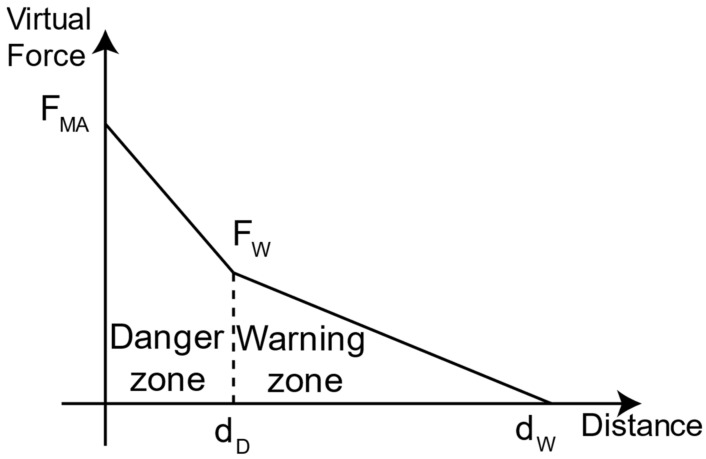
Virtual force generated by an obstacle at a given distance in the environment. The force is proportional to the distance, with different stiffness constants, based on the zone (i.e., “warning” and “danger” zones).

**Figure 4 sensors-23-00677-f004:**
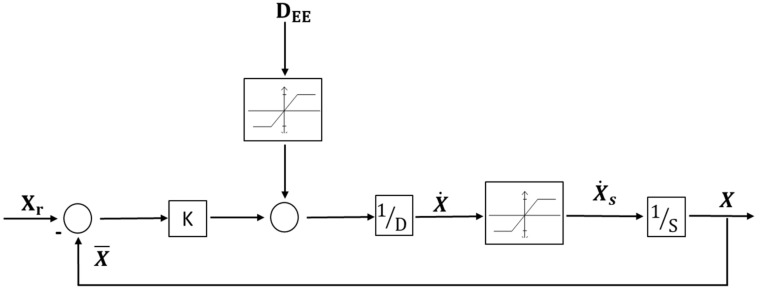
Schematic of the SISO linear system which controls the dynamic evolution of each state variable. The inputs are the reference state vector and the disturbance vector. The saturation for state velocity and disturbance is also reported.

**Figure 5 sensors-23-00677-f005:**
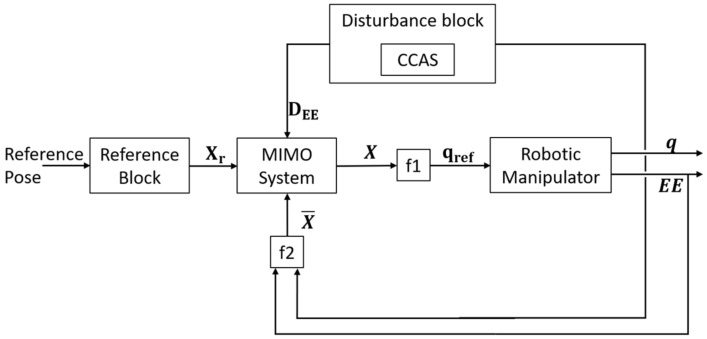
General architecture of the local path planner.

**Figure 6 sensors-23-00677-f006:**
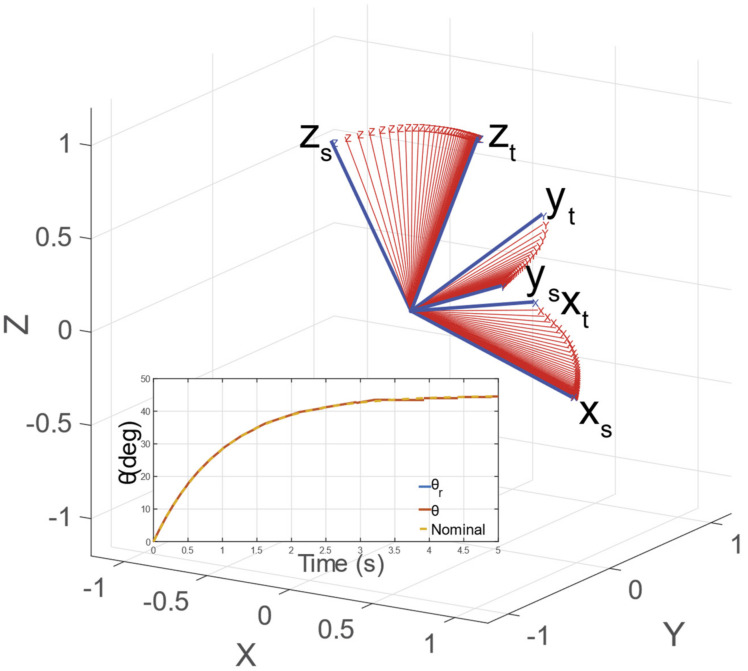
Angular motion from an initial to the desired orientation of the manipulator’s EE (depicted in blue). In red, the time evolution of the EE orientation sampled at 0.1 s is reported. The inset represents the θ dynamical response for the depicted angular motion.

**Figure 7 sensors-23-00677-f007:**
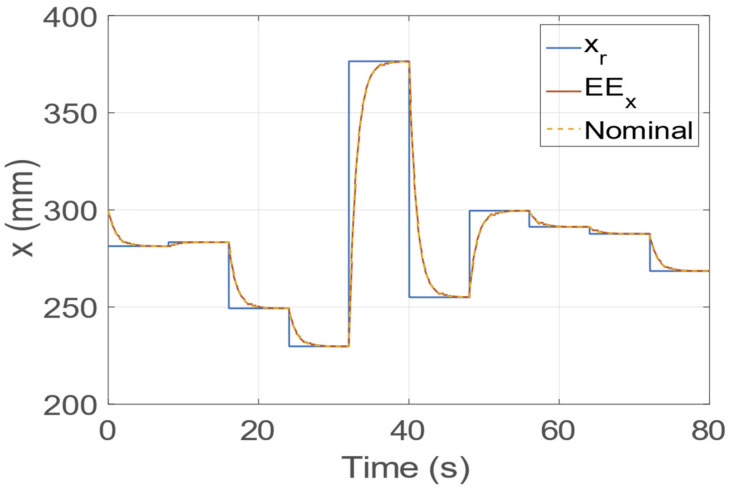
Dynamic evolution of the x coordinate with the relative reference and nominal signals.

**Figure 8 sensors-23-00677-f008:**
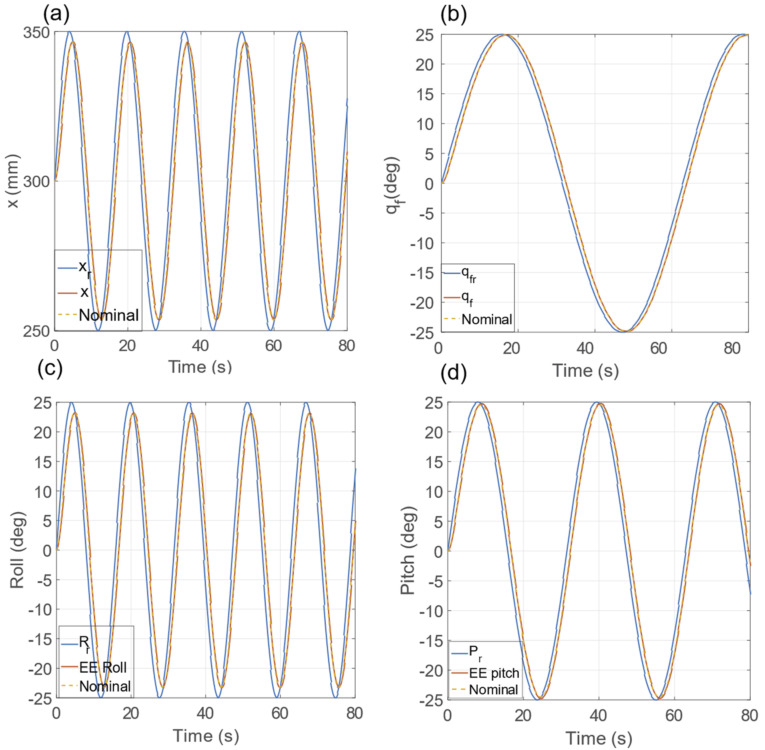
Dynamic evolution of x (**a**), q_t_ (**b**), roll (**c**), and pitch (**d**) variables compared with the nominal dynamics and the relative reference signal for the sinusoidal input.

**Figure 9 sensors-23-00677-f009:**
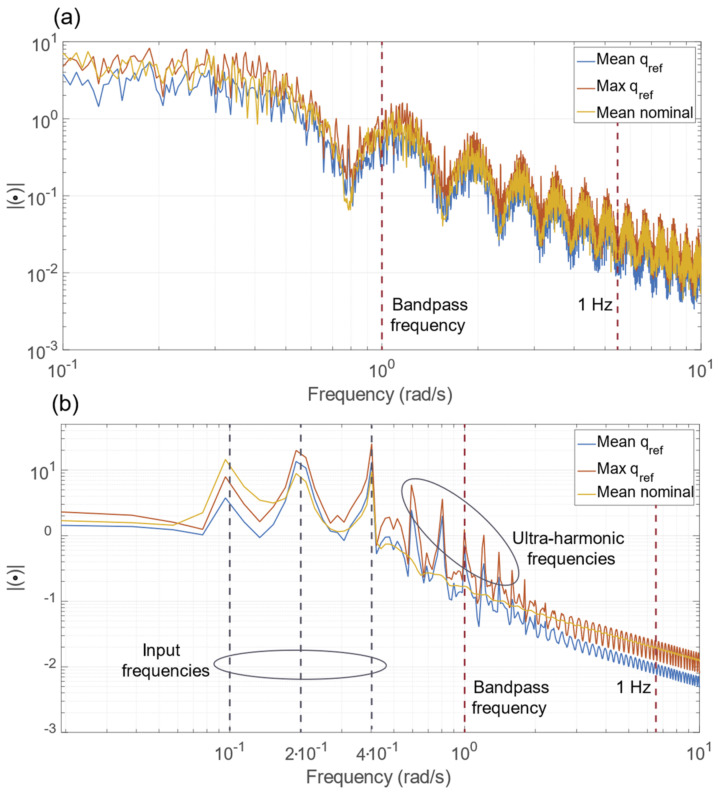
(**a**) Frequency analysis for the step path. The spectra of the mean and the maximum q_ref_ signal are presented, compared with the spectrum of the nominal dynamics highlighting the band-pass frequency of F(s). (**b**) Frequency analysis for the sinusoidal path. The spectra of the mean and the maximum q_ref_ signal are depicted, compared with the spectrum of the nominal dynamics highlighting some notable frequencies.

**Figure 10 sensors-23-00677-f010:**
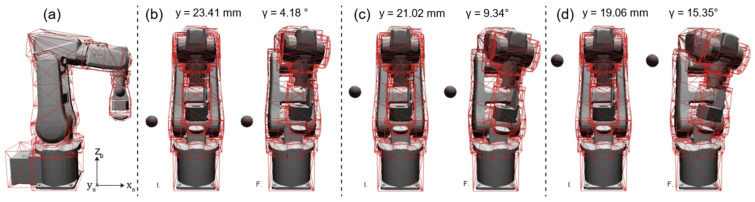
(**a**) Lateral view of the robotic manipulator with one obstacle. The direction of the base reference system axes is also reported. (**b**) The initial (I.) and final (F.) pose of the manipulator when the obstacle is placed at 350 mm on the z_B_ coordinate. (**c**) The initial and final pose of the manipulator when the obstacle is placed at 450 mm on the z_B_ coordinate. (**d**) The initial and final pose of the manipulator when the obstacle is placed at 550 mm on the z_B_ coordinate.

**Figure 11 sensors-23-00677-f011:**
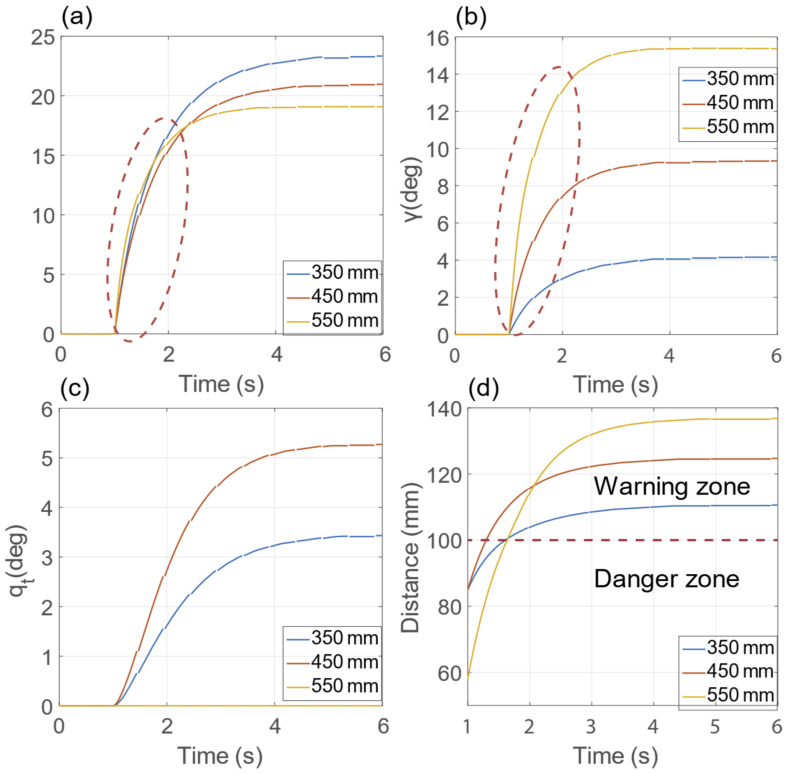
Dynamic evolution of the y (**a**), γ (**b**), q_t_ (**c**) state variables and time response (**d**) of the minimum distance for the three performed simulations with one obstacle.

**Figure 12 sensors-23-00677-f012:**
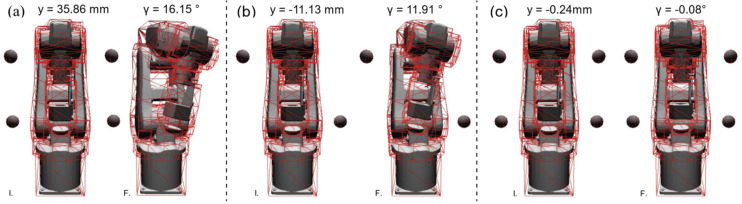
(**a**) Initial and final pose of the manipulator when two obstacles are placed on the same manipulator side. (**b**) The initial (I.) and final (F.) pose of the manipulator when two obstacles are placed on different manipulator sides. (**c**) The initial and final pose of the manipulator with four symmetric obstacles.

## Data Availability

Starting from random initial conditions/constraints (i.e., obstacle position, number of obstacle), as described in Section 2.2.1 and Section 2.2.2, data about the relative distances between the robot manipulator and obstacles were computed by our CPS software [20]; the model presented in Section 2.1.3 was then used to compute the robot trajectories in the cartesian space.

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
