# Peer review of "Spherical Wrist Manipulator Local Planner for Redundant Tasks in Collaborative Environments"

_sensors, 2023, doi:10.3390/s23020677_

Round 1

Reviewer 1 Report

The paper presents the implementation of a control policy for obstacle avoidance that takes advantage of the redundancy of a serial robot. The topic is interesting and falls within the important research trend of redundant robotics. However, several concerns should be addressed before the paper is ready for publication.

  ** Section 2.1.1. is rather confusing, mostly because of the lack of formal definition of the axes and frames used. The difference between z_t, z_s and z^{\theta}_{EE} should be clarified. It is the reviewer's belief that the orientation of the end-effector can be parametrized in a much simpler way without having to define this number of axes and frames. ** The authors constantly misuse the word 'versor' to refer to unit vectors. Note that a versor commonly refers to a unit quaternion. ** The difference between \gamma_r, \overline{gamma} and \gamma should be clarified. Similarly, \theta and \theta_r are not clearly defined. The improvement of Fig. 2 would help these definitions - use hidden-line removal or any other graphic technique that can give a more tridimensional feel. ** The notation throughout the paper should be improved. In some cases, like d_D in Eq. (7), the variable is written in bold while clearly such a variable is a scalar. In Eq. (9), for example, even the scalar 1 is bold-faced. ** The definition of torques as well as why they are needed should be clarified. The description in lines 186 and 187 is not a correct definition. ** The purpose of transformation in Eq. (12) should be stated. The transformation matrix expressed in brackets seems to be the inverse (transpose) of the orientation of frame EE with respect to world frame. Is T^s_{EE} the torque T_{EE} in EE frame? Due to the use of multiple frames, the notation of each vector should specify the frame it is referred to. ** Note that D_{EE} in Eq. (13) combines torque and force units and, hence, problems related to ill-conditioning may appear. The authors should explain why the use of such a matrix does not lead to problems of this kind.

** A more detailed literature research must be added in order to see where this paper is placed in the current state of the research on redundant robotics. There are many papers on this topic and important publications that tackle closely related problems are not cited.

Reviewer 2 Report

Aiming at the redundant situation in production and life, this article proposes a new planning method. The content of the article is substantial and the research significance is clear. But I think there are still some problems.

1.      In the article, only the first sentence of the first paragraph of the introduction part has a blank space, and the rest of the full-text paragraphs are written in the top grid. Authors are requested to write according to the journal template.

2.      The spacing in the article is inconsistent, such as the spacing of lines 231-236 is obviously inconsistent with other parts.

3.      The formulas in the full text are bolded in black italics. Generally speaking, scalars and symbols are in italics. Please refer to the template and published papers to modify the writing of formulas and symbols.

4.      The manipulator model shown in the picture is too abstract, please add a complete schematic diagram of the manipulator.

5.      Comparison need to be carried out with existing techniques if possible compare it with recent trends.

6.      Try to include and refer recent papers in references and mention their contribution in your Literature Survey

Round 2

Reviewer 2 Report

The revised version uploaded by the author has traces of comments. The author only needs to color in the revised parts, and there is no need to leave a comment trail.